# Chiral active particles are sensitive reporters to environmental geometry

Chung Wing Chan[1], Daihui Wu[1], Kaiyao Qiao [1], Kin Long Fong[1,2], Zhiyu Yang[1], Yilong Han [1] & Rui Zhang [1] ✉

Chiral active particles (CAPs) are self-propelling particles that break time-reversal symmetry by orbiting or spinning, leading to intriguing behaviors. Here, we examined the dynamics of CAPs moving in 2D lattices of disk obstacles through active Brownian dynamics simulations and granular experiments with grass seeds. We find that the effective diffusivity of the CAPs is sensitive to the structure of the obstacle lattice, a feature absent in achiral active particles. We further studied the transport of CAPs in obstacle arrays under an external field and found a reentrant directional locking effect, which can be used to sort CAPs with different activities. Finally, we demonstrated that parallelogram lattices of obstacles without mirror symmetry can separate clockwise and counter-clockwise CAPs. The mechanisms of the above three novel phenomena are qualitatively explained. As such, our work provides a basis for designing chirality-based tools for single-cell diagnosis and separation, and active particle-based environmental sensors.

Active matter represents a diverse range of non-equilibrium systems, in which their constituents can autonomously move or spin by converting energy into mechanical work; this phenomenon leads to intriguing collective dynamics, such as flocking[1–4], activity-induced phase separation[5–8], emergent hyperuniform structures[9], and spontaneous flows[10,11]. Recent research interests in active matter are mainly motivated by its emerging phenomena that do not exist in equilibrium systems, its biological relevance[12–14], and its potential applications in autonomous materials systems[15–19]. Our current understanding of active matter is still limited because of its far-from-equilibrium nature[20,21].

There are two types of active matter systems, namely wet and dry active matter. In wet systems, particles, droplets, or living matter self-propel in a liquid[22–25]. Small robots are a typical example of dry systems, which offers valuable insights into the collective behaviors of active matter in response to their environments. Robot swarms, which navigate, sense, and interact with their environments, demonstrate morphogenesis and on-demand reconfiguration to perform various functions against their surroundings[26–29]. Magnetic fields have been widely adopted as external actuation sources for robot swarms, enabling wireless actuation in complex bio-fluids and showcasing

significant potential in biomedical applications like targeted drug delivery[30–32]. Moreover, these active robot swarms can effectively mimic collective behaviors observed in ecological systems, therefore enabling physical modeling of evolving systems[33].

Another common type of dry active matter system is granular particles that can self-propel or self-spin under an energy supply, such as on a vibrating substrate[34,35]. Active matter can also be categorized into linear and rotational self-propelling particles.

Recent research has been focused on linear active particles in complex environments due to their mesmerizing transport properties[36–40]. Linear active particles tend to be trapped by boundary walls, a feature not found for passive particles at thermodynamic equilibrium[41,42]. Moreover, linear active particles exhibit directional locking effect[43–46] and topotaxis in obstacle arrays[47], and active ratchet effects under asymmetric boundaries[48–51].

The less studied rotational particles exhibit kinetic chirality and are often called chiral active particles (CAPs)[33,52–57]. Kinetic chirality at the single-particle level originates from the particle's asymmetry, such as surface coating, body shape, or mass distribution[58–62]. These systems can lead to intriguing phenomena, including edge currents and odd viscosities[63–67].

[1]Department of Physics, The Hong Kong University of Science and Technology, Clear Water Bay, Kowloon, Hong Kong SAR. [2]Present address: Physik-Department, Technische Universität München, James-Franck-Straße 1, 85748 Garching, Germany. ✉e-mail: ruizhang@ust.hk

CAPs in complex environments, such as a lattice of obstacles, exhibit fascinating transport properties due to the interactions between active particles and obstacles[68–72]. Simulations show that the effective diffusivity of active particles is not necessarily slower in a crowded environment[73,74]. Bacteria in colloidal suspensions or polymer solution display similar mobility enhancement behavior[75–77].

Many biological systems are often intrinsically chiral. Experiments on chiral active matter interacting with complex environments are mainly focused on living systems such as bacteria[78–81]. Our current understanding of these chiral active entities transport in complex biological environment is overwhelmed by the intricacy of the system involving unclear physical and biochemical factors[55,65,82]. In addition, there is a recent interest in active spinner systems due to their odd viscosities and topological edge currents[63–65]. However, it remains unclear how this type of active matter is related to other active systems such as the more commonly studied linear active matter.

In this work, we introduce a new type of granular CAPs, namely, grass seeds, to tackle the above scientific questions. These seed particles are smaller and lighter than previous man-made granular CAPs and are thus suitable for studying CAP motions in obstacle arrays. We investigate how CAPs are transported in obstacle arrays through active Brownian dynamics simulations and granular experiments. Our work provides a simple and convenient platform to study the interplay between chiral active matter and complex environments. The CAPs considered in this work combine the properties of linear propulsion and self-spinning, thus serving as a flexible system to bridge two distinct active matter systems, namely linear and chiral active matter. In contrast to existing works that focus on separating particles with different chiralities using a specific obstacle lattice[83–86], here, we vary lattice parameters and particle chirality and observe novel effects in three systems: (1) abnormal diffusion in periodic lattices, (2) chirality-mediated directional locking in a periodic lattice with an external field, and (3) effective diffusivity difference in lattices without mirror symmetry. Our work reveals that chiral active matter is sensitive to the environmental geometry, including obstacle lattice configuration and the degree of lattice asymmetry. Beyond the commonly discussed applications of active matter in separation and therapeutic delivery, our work also paves the way toward its novel applications such as using chiral active matter as environmental sensors.

## Results

### Chiral active particle (CAP) model

The CAPs are modeled as overdamped active Brownian disks with radius $R_p$, self-propelled at a linear velocity $v_0$ and angular velocity $\omega_0$ along a time-dependent orientation angle $\theta(t)$ with respect to the $x$-axis. The equations of motion for a CAP are as follows[2,45,47,87]:

$$\frac{d\vec{r}}{dt} = v_0 \hat{p} + \alpha \vec{F}, \tag{1a}$$

$$\frac{d\theta}{dt} = \omega_0 + \sqrt{2D_r}\xi(t), \tag{1b}$$

where $\xi(t)$ is a rotational white noise with zero mean, $\langle\xi(t)\rangle = 0$ and $\langle\xi(t)\xi(t')\rangle = \delta(t - t')$. The unit vector $\hat{p} = (\cos\theta, \sin\theta)$ rotates stochastically with a rotational diffusion coefficient $D_r$. Here we only consider rotational noise since it has more pronounced effect on particle trajectories than linear velocity noise[87]. The mobility $\alpha$ and force $\vec{F}$ embody the interaction between the particle and the obstacles. Note that the vertical collisions between the CAP and the substrate give rise to an effective active Brownian motion like behavior for the CAP in the horizontal plane, the inertial effect is accounted for by the model parameters $v_0$, $\omega_0$, and $D_r$.

In free space, $\vec{F} = 0$ and the particle follows a circular trajectory of orbital radius $r = v_0/\omega_0$ (Fig. 1a). If $\omega_0 = 0$, the CAPs would reduce

to achiral active Brownian particles (ABPs), whose motions can be described by a single length scale, namely persistence length $l_p = v_0\tau_p$, which is a typical distance traveled by a particle before it changes its orientation. The persistence time $\tau_p = 1/D_r$ describes the time for the particle to forget its initial orientation[2]. For CAPs with $\omega_0 \neq 0$, diffusivity in free-space can be analytically solved as follows[52,87]:

$$D = \frac{v_0^2}{2\omega_0}\frac{\Omega}{1+\Omega^2}, \tag{2}$$

where $\Omega = D_r/\omega_0 = r/l_p = \tau_\omega/\tau_p$ characterizes the system as dominated by circular motion ($\Omega \ll 1$) or noise ($\Omega \gg 1$, Fig. S1a). $\tau_\omega = 1/\omega_0$ is the time for a CAP to complete one cycle of revolution. Our simulation confirms Eq. (2) (Fig. S1b, c), wherein $D$ is maximized at $\Omega = 1$ (Fig. S1d), that, $\tau_\omega = \tau_p$.

We simulate CAPs in 2D square and triangular lattices of circular obstacles of radius $R_o$. The force between the particle and obstacle $j$ centered at $\vec{r}_j$ is[47]:

$$\vec{F}_j = \begin{cases} -\frac{v_0}{\alpha}\left(\hat{p}\cdot\hat{N}_j\right)\hat{N}_j, & \text{if } |\vec{r}-\vec{r}_j| \leq R, \\ 0, & \text{otherwise ,} \end{cases} \tag{3}$$

where the unit vector $\hat{N}_j$ is normal to the obstacle boundary at the collision point[42]. The normal component of the velocity that would drive the particles into the wall is canceled by the wall force. This wall potential does not depend on the specific value of $\alpha$, which is only retained for dimensional consistency[42]. The particle radius $R_p = 0$ is used in the simulation. In the experiment, the finite particle radius $R_p$ causes a minimum separation between a particle and an obstacle to be $R = R_p + R_o$. Therefore, we take the effective radius of the obstacle as $R$ in calculating the effective area fraction $\phi$ of the obstacle lattice. The grass seed is regarded as a point particle with $R_p = 0$ in the simulation. $\phi$ is fixed when the dynamics of CAPs are compared between different lattices. We use the particle's self-propelling velocity $v_0$ and the effective radius $R$ as the basic units and set $v_0 = R = 1$ with the time scale being $R/v_0$. Other physical quantities are rescaled by these units, i.e., $\bar{l}_p = l_p/R, \bar{r} = r/R, \bar{\omega}_0 = 1/\bar{r}$, and $\bar{D}_r = 1/\bar{l}_p$. Noted that the simulation model can be fully described by the two parameters, namely, $\bar{l}_p$ and $\bar{r}$.

**Echinochloa crus-galli' seed as CAP.** Granular particles are often driven by a vibration stage and exhibit random motions. Among the seeds of various plants, only the seeds of *Echinochloa crus-galli* grass (Figs. 1c, d and S2) exhibit circular motions on a vibration stage (Fig. 1d) or in the sound wave of human's voice. Such circular motion arises from the protrusion grooves on the seed surface. The circular motion vanishes when the ridged skin is removed (Figs. 1e and S2).

Different seeds have slightly different surface grooves, resulting in different modes of motion, including spinning, random motion, and circular locomotion, on a vibrating substrate. We use seeds 1 and 2, which persistently orbit in clockwise and counterclockwise directions, respectively, at constant angular speeds (Fig. 1f, Table 1). The typical trajectories are shown in Fig. 1a and Supplementary Videos 1 and 2. The velocity is always along the long axis of the seed (Fig. S5). We also fabricate 3D plastic particles with three surface grooves by 3D printing (Fig. S2), but they are heavier than the seeds and thus exhibit lower $\bar{r}$ (Supplementary Video 3). To achieve high motility and large $\bar{r}$, we use grass seeds as CAPs in our experiment.

### Abnormal diffusion

We observe two types of abnormal diffusion by systematically tuning the orbital radius $\bar{r}$ and the obstacle packing fraction $\phi$ of the lattice. A CAP with $\bar{r} \approx 1$ is mainly caged in the square lattice (Fig. 2a) but diffuses rapidly in the triangular lattice at high $\bar{l}_p$ with the same $\phi = 0.7$ (Fig. 2b). By contrast, a CAP with $\bar{r} \approx 2$ exhibits the opposite behavior in these

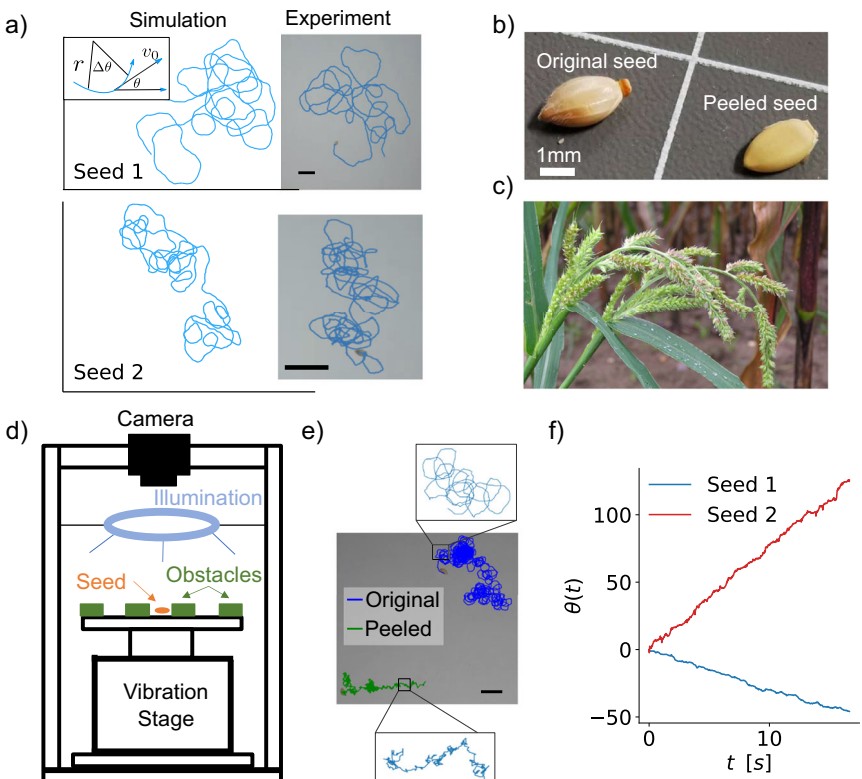

**Fig. 1 | Granular CAPs. a** Simulated trajectories of CAPs in free space match with the experimental trajectories of two seeds with different orbital radii. Seed 1 (top) with a large $r \approx 15$ mm. Seed 2 (bottom) with a small $r \approx 1.6$ mm (Table 1). Inset in (**a**) illustrates the definitions of $r$, $\theta$, $\Delta\theta$, and $v_0$. **b** Original seed with surface ridges (left) and peeled seed without ridges (right). **c** *Echinochloa crus-galli* grasses with seeds (taken by Michael Becker, CC BY-SA 3.0[97]). **d** Experimental setup. **e** Circular trajectory (blue) exhibited by the original seed, and random non-circular trajectory (green) exhibited by the peeled seed in free space. **f** Measured seed orientation $\theta$ as a function of time $t$ for the two seeds showing that their angular velocities $\omega_0$ are constants. Scale bars in (**a**, **e**): 1 cm.

**Table 1 | Model parameters for grass seeds 1 and 2 measured in experiments and in simulations**

| | | Experiment | | | Simulation | | |
|---|---|---|---|---|---|---|---|
| | | Units | Seed 1 | Seed 2 | | Seed 1 | Seed 2 |
| Effective radius | $R$ | (mm) | 7.0 ± 1 | 7.0 ± 1 | $R$ | 1.0 | 1.0 |
| Self-propelling linear velocity | $v_0$ | (mm s$^{-1}$) | 39 ± 1 | 9.7 ± 0.1 | $v_0$ | 1.0 | 1.0 |
| Orbital radius | $r$ | (mm) | 15 ± 2 | 1.6 ± 0.1 | $\bar{r} = r/R$ | 1.7 | 0.2 |
| Persistence length | $l_p$ | (mm) | 25 ± 2 | 3.6 ± 0.2 | $\bar{l}_p = l_p/R$ | 3.2 | 0.5 |
| Angular velocity | $\omega_0$ | (s$^{-1}$) | −2.6 ± 0.4 | 6.3 ± 0.3 | $\bar{\omega}_0 = 1/\bar{r}$ | 0.59 | 5.0 |
| Orientational diffusion coefficient | $D_r$ | (s$^{-1}$) | 1.6 ± 0.1 | 2.7 ± 0.2 | $\bar{D}_r = 1/\bar{l}_p$ | 0.31 | 2.0 |

The values in simulation units are adopted for comparison with the experimental trajectories in Fig. 1a.

lattices. These types of behavior are confirmed in the experiment (Figs. 2c, d and S5 and Supplementary Videos 4–7).

The motions of CAPs are quantified by mean square displacement (MSD) $\langle |\Delta\vec{r}|^2 \rangle$. We find that MSD $\propto t^2$ (i.e., ballistic) at short times and $\propto t$ (i.e., diffusive) at long times (Fig. 3). The result is in accordance with our expectation that a CAP moves ballistically at $t \ll \tau_p$ and diffuses randomly at $t \gg \tau_p$ because it orbits many cycles during a long time step. Therefore, the effective diffusion constant (diffusivity for short) $D_{\text{eff}} = \langle |\Delta\vec{r}|^2 \rangle / 4t$ for CAPs can be measured in the long-time limit[2,10,37,47]. It is believed that the diffusion of active particles in an obstacle lattice is always slower than that in free space due to collision and clogging by obstacles[47,88]. However, our simulations and experiments show that certain obstacle lattices can enhance or suppress CAP diffusion under different circumstances (Figs. 4 and 5).

We systematically measure $D_{\text{eff}}$ in a broad range of $\bar{l}_p \in [10^0, 10^3]$ and $\bar{r} \in [0, 8]$ in free space, square lattices, and triangular lattices. The ratios to $D$ in free space spans eight orders of magnitude; thus, we plot the logarithm of their ratios, $\Psi = \ln\frac{D_{\text{eff}}}{D}$, in Figs. 4a, b and S6. Figure 4a, b shows that the diffusivity of CAPs behaves oppositely in the square and triangular lattices with the same area fraction. For instance, at high $\bar{l}_p$ ($\geq 100$), CAPs diffuse faster in a dense square lattice than in free space (i.e., $\Psi > 0$) when $\bar{r} \approx 2, 4$, but less mobile ($\Psi < 0$) at $\bar{r} \approx 1, 3, 5$ (Fig. 4a). By contrast, $D_{\text{eff}} > D$ when $\bar{r} \approx 1, 3, 5$ and $< D$ when $\bar{r} \approx 2, 4$ in the triangular lattice (Fig. 4b). The oscillatory behavior of $\Psi$ with respect to $\bar{r}$ persists up to $\bar{r} \approx 5$, beyond which $\Psi$ becomes insensitive to $\bar{r}$ (Fig. 4c). This chirality-sensitive behavior of diffusivity also diminishes for small $\bar{l}_p$ at which active particles quickly lose their directionality (Fig. 4d). In general, the relative diffusivity parameter $\Psi$ increases monotonically as $\bar{l}_p$ increases, because higher activity can enhance diffusivity (Fig. 4a, b). However, a special case in which $\Psi(\bar{l}_p)$ is non-monotonic occurs for the square lattice at $\bar{r} = 1$ (Fig. 4d).

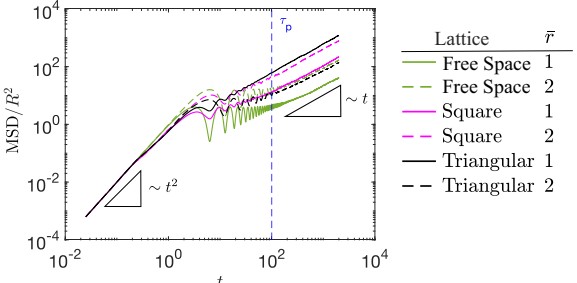

**Fig. 2 | Trajectories of CAPs in obstacle lattices.** Typical simulation trajectories of CAPs at high persistence length $\bar{l}_p = 100$ with $\bar{r} = 1$ (green) and $\bar{r} = 2$ (purple) in square lattice (**a**) and triangular lattice (**b**) with the same packing fraction $\phi = 0.7$.

Experimental trajectories of seed 1 (green) and seed 2 (purple) in square lattice (**c**) and triangular lattice (**d**). Scale bar: 1 cm.

**Fig. 3 | MSDs of CAPs in free space and obstacle lattices.** Comparison of MSD of CAP with $\bar{l}_p = 10^2$ and $\bar{r} = 1,2$. MSD $\propto t^2$ at $t \ll \tau_p$ and $\propto t$ at $t \gg \tau_p$. The packing fraction $\phi = 0.7$ for all the obstacle lattices considered here. The dashed blue line indicates the $\tau_p$ of CAPs in free space.

Diffusivity depends not only on $\bar{r}$ and $\bar{l}_p$ (Fig. 4) but also on the packing fraction $\phi$ of the lattice (Fig. 5a, b). In sparse lattice ($\phi \lesssim 0.35$), particle diffusivity lowers as $\phi$ increases, consistent with one's intuition. At high packing fractions ($\phi \gtrsim 0.35$), however, $D_{\text{eff}}$ drastically increases as $\phi$ increases. We further compare the diffusion behavior of CAPs with ABPs. Figure 5c shows no significant difference in $D_{\text{eff}}(\phi)$ for ABPs diffusing in the two types of lattices, in contrast to the distinct diffusion of CAPs. Moreover, $D_{\text{eff}}(\phi)$ monotonically decreases for ABPs due to stronger caging at higher $\phi$ (Fig. 5c). By contrast, CAPs can diffuse faster in lattices with higher $\phi$ (Fig. 5a, b).

Note that when the obstacle lattice is imperfect, interesting transport phenomena may emerge. For example, if an obstacle lattice consists of two different lattices with different diffusivities, particles will exhibit topotaxis effect by migrating into lower-diffusivity region[10,47]. Our additional simulations demonstrate that topotaxis effect is also present for CAPs moving in a binary lattice with identical packing fraction (Fig. S7). If a single lattice of obstacles is subject to a noisy configuration, CAP diffusivity will deviate from that of the reference perfect lattice; more details are provided in SI (Fig. S8).

To understand the mechanisms of fast and slow diffusion, we further examine the motion of CAPs on obstacle surfaces. Active particles including CAPs and ABPs orbit along the surface of an obstacle and frequently hop to a neighboring obstacle. We find that CAPs occasionally reverse their motion direction on obstacle surfaces, which is rarely seen in ABPs. When the CAP pushes against the obstacle, it moves along the surface of the obstacle; otherwise, it leaves the surface. Consequently, it always leaves the obstacle along the tangential direction (Fig. 6). The moving direction of a CAP along the surface of the obstacle is dictated by the tangential component of the intrinsic orientation, which is constantly rotating. The two directions always make an acute angle when the CAP is gliding on the obstacle (Fig. 6a). When the tangential component of the intrinsic orientation changes direction at $t_1$ in Fig. 6a, the moving direction reverses.

To understand the abnormal diffusivity we have revealed (Figs. 4 and 5), we measure the number of the reversible motion $N$ and the ratio

of the sliding time along the obstacle surface to hopping time between obstacles $\mu = t_{\text{slide}}/t_{\text{hop}}$ in Fig. 6. Specifically, for a fixed duration of simulation time $t_{\text{total}}$, we measure the sliding time, $t_{\text{slide}}$, defined as the cumulative time when CAPs are in contact with the surface of obstacles (i.e., there exists an obstacle $i$ such that $|\vec{r} - \vec{r}_i| \leq R$). Particle hopping time is then defined via $t_{\text{hop}} = t_{\text{total}} - t_{\text{slide}}$. During sliding, we count the number of occurrences of reversible motion of the particle as $N$ (i.e., particle's tangential velocity with respect to the obstacle surface normal changes sign). Both parameters we have measured show strong positive correlations with $\Psi$, indicating that they dominate the diffusion constant $D_{\text{eff}}$ (Fig. 6).

To understand, note that the diffusion of CAPs is primarily driven by their ability to hop to the next row of obstacles, which effectively initiates a new orbit. The occurrence of a larger number of reversible motions on the surface of obstacles provides CAPs with more opportunities to explore the surrounding space. Consequently, this increased exploration enhances the likelihood of CAPs successfully hopping to the next row of obstacles. Notably, our results demonstrate a strong positive correlation between $N$ and the effective diffusivity of CAPs.

A larger $\omega_0$ (small $\bar{r}$) produces a shorter $t_2$ (Fig. 6), that is, shorter $t_{\text{slide}}$, corresponding to larger $\mu$ and $N$, and more hopping. More hopping usually produces a faster diffusion except that the trajectory forms a nearly closed loop, see the strongly caged motion in Fig. S9b. These abnormal diffusions demonstrate that CAPs are sensitive reporters to environmental geometry.

## Chirality-mediated directional locking

We further study the transport of CAPs subjected to an external field in obstacle lattices. When traveling through a periodic obstacle lattice subjected to a global flow, particles often-times experience a so-called "directional locking effect", i.e., the particles are locked to certain mean migration directions[89,90]. It is known that anisotropic-shaped obstacles can lead to controlled directional migration of microswimmers[39,43]. Here we focus on easy-to-fabricate circular disks and find a chirality-induced reentrant directional locking effect at high activities (i.e., large $\bar{l}_p$) as elaborated in the following.

Directional locking for CAPs in circular obstacles is studied against directional locking for linear active particles or anisotropic obstacles[44–46]. In the simulation, we impose a global flow with speed $v_g$ to the particles. Thus, Eq. (1a) is modified as follows[44,90]:

$$\frac{d\vec{r}}{dt} = v_0 \hat{p} + v_g \hat{e} + \vec{F}, \tag{4}$$

where the unit vector $\hat{e} = (\cos \psi, \sin \psi)$ represents the direction of the global flow, with $\psi$ as the angle between the flow direction and the $x$-axis. We fix the global flow strength at $v_g/v_0 = 1$ and vary its direction from $\psi = 0°$ to $90°$ for ABPs and CAPs with $\bar{r} = 1$. Figure 7a–d shows the mean migration direction $\alpha = \tan^{-1}(\langle v_y \rangle / \langle v_x \rangle)$ versus the global flow angle $\psi$. The measured $v_i = (r_i(t + \Delta t) - r_i(t))/\Delta t$ for $i = x, y$[44,90]. The plateaus in Fig. 7a–d represent the locking direction and deviation from the global flow.

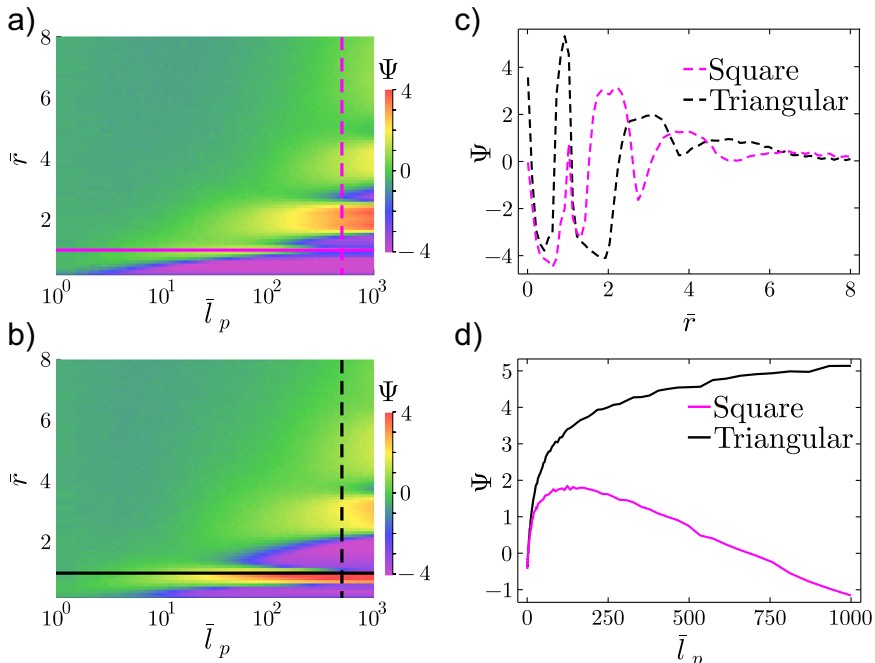

**Fig. 4 | Comparisons of effective diffusivity.** Logarithmic ratio of effective $D_{eff}$ to free-space diffusivity $D$, namely $\Psi = \ln\frac{D_{eff}}{D}$ for square (**a**) and triangular (**b**) lattices at the same packing fraction $\phi = 0.7$. Red and purple regions represent $D_{eff} > D$ and $D_{eff} < D$, respectively. $\Psi$ at $\bar{l}_p = 500$ (**c**) and $\bar{r} = 1$ (**d**), corresponding to vertical dashed lines and horizontal solid lines in (**a**, **b**), respectively.

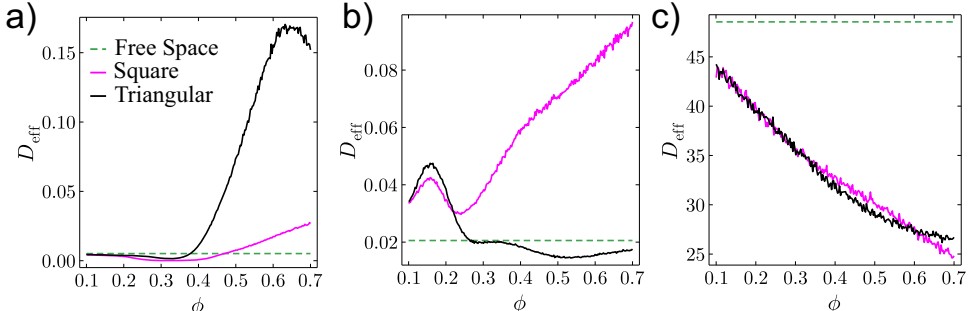

**Fig. 5 | Effects of packing fraction on effective diffusivity.** $D_{eff}$ as a function of obstacle packing fraction $\phi$ for CAPs at $\bar{r} = 1$ (**a**), CAPs at $\bar{r} = 2$ (**b**) and ABPs (**c**). All persistence lengths are chosen to be $\bar{l}_p = 100$.

As shown in Fig. 7a, b, CAPs and ABPs exhibit strong locking effect and identical behavior at small $\bar{l}_p$. In the passive Brownian particle limit, the locking directions are along the symmetry axes with 0°, 45°, 90° for the square lattice and 0°, 60° for the triangular lattice[44]. However, in the large $\bar{l}_p$ regime, the locking effect is only found for CAPs (Fig. 7c, d) and such chirality-mediated locking directions are no longer along the symmetry axes of the lattice (Fig. 7e, f). The most robust locking directions are 65° for the square lattice and 40° for the triangular lattice (Fig. 7c, d).

We quantify the strength of the locking effect by:

$$\epsilon = \int_0^{\pi/2} (\alpha - \psi)^2 d\psi \qquad (5)$$

from the $\alpha-\psi$ graph in Fig. 7a–d. For ABPs, $\alpha$ is only locked at small $\bar{l}_p$, and the locking effect disappears when $\bar{l}_p \geq 1$ (Fig. 8), consistent with ref. 44. The directional locking effect is weakened for both types of active particles as $\bar{l}_p$ increases and completely vanishes at $\bar{l}_p = 1$. As $\bar{l}_p$ is further increased to more than 1, the directional locking reappears for

CAPs in both lattices. The reentrance of the directional locking effect is absent for ABPs. Specifically, the directional locking effect only appears in the passive Brownian particle limit for ABPs (Fig. S11). We also investigate the directional locking effect at different $\bar{r}$ values and found that it is weaker at larger $\bar{r}$.

The trajectories of CAPs exhibit zigzag patterns (Fig. 7e, f), which are also observed in the passive non-Brownian particles driven in microfluidic systems[46,89] and driven skyrmion systems[90]. The zigzag pattern reflects the fact that the particle hops to the neighboring column by a certain arc periodically. For CAPs considered here, they tend to keep the steady change of the angular orientation. When an external driving force enhances the locomotion of CAPs in a specific direction, the lattice structure scatters the intrinsic circular motion of the particles, resulting in a zigzag motion along a new direction. Note that the directional locking effect can be used as a candidate mechanism to design a mixed obstacle lattice to guide a CAP along a specific trajectory. However, due to the presence of diffusion noise, the success of this task will become increasingly challenging as the prescribed trajectory lengthens.

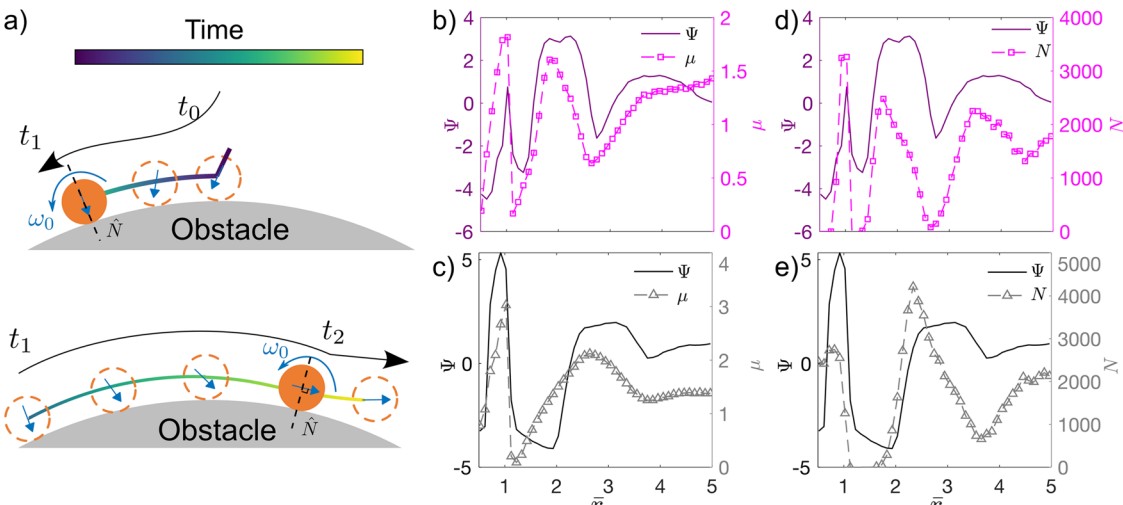

**Fig. 6 | Reverse and hopping motions of CAPs. a** Particle motions along the obstacle surface. The intrinsic orientation $\hat{p}$ (blue arrows) of the particle rotates at $\omega_0$. After the particle lands on an obstacle at $t_0$, the direction of blue arrows dictate the moving direction. The tangential moving direction (black arrows) is along the tangential component of blue arrows; thus, the moving direction reverses when the blue arrow becomes normal to the obstacle surface at $t_1$. When the blue arrow is parallel to the obstacle surface at $t_2$ (i.e., the normal component of the blue arrow is not against the obstacle), the particle leaves the obstacle. $\Psi = \ln \frac{D_{\text{eff}}}{D}$ and the ratio of hopping time $\mu = t_{\text{slide}}/t_{\text{hop}}$ in square (**b**) and triangular (**c**) lattice. Relation between the logarithmic ratio of diffusivity $\Psi$ and the number of reversible motions $N$ in square (**d**) and triangular (**e**) lattices.

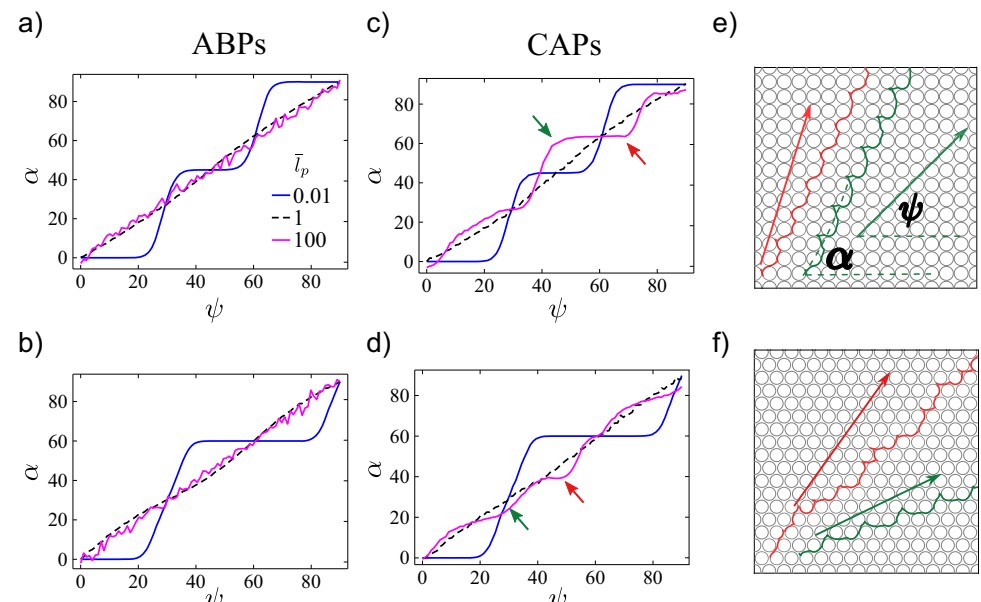

**Fig. 7 | Directional locking effects for ABPs and CAPs.** Mean particle migration angles $\alpha$ against global flow angles $\psi$ in square (**a**, **b**) and triangular (**c**, **d**) lattices with $\bar{l}_p = 0.1, 1$ and 100. **e** Trajectories of CAPs in a square lattice with the global flow at $\psi = 45°$ (green) and $\psi = 68°$ (red). **f** Trajectories of CAPs in a triangular lattice with the global flow at $\psi = 28°$ (green) and $\psi = 50°$ (red). Arrows in (**e**) and (**f**) indicate the direction of global flow $\psi$; corresponding angles are pointed in (**c**) and (**d**) by small arrows with the same color. We fixed the system at $\phi = 0.7$ and $v_g/v_0 = 1$, and $\bar{r} = 1$ for CAPs.

## Sorting CAPs in lattices with symmetry breaking

The two types of CAPs with opposite handedness are clockwise (CW) and counter-clockwise (CCW) particles. Their motions cannot be distinguished in square or triangular lattices with mirror symmetry. Existing simulations have considered difficult-to-fabricate non-circular obstacles to separate CW and CCW particles[83,91]. Here we propose using circular obstacles to form a simple chiral lattice to separate CW and CCW particles.

Specifically, we consider a mirror-symmetry broken parallelogram lattice (Fig. 9a–d), which can be created by horizontally shifting each layer of a square lattice by a constant distance $\delta d$, where $d$ is the lattice constant and $\delta$ is the shape parameter with $\delta \in [0, 1)$ (Fig. 9e). The packing fraction $\phi$ of the lattice will be invariant during this operation. The deformed lattice preserves the mirror symmetry at $\delta = 0, 0.5$ and 1 (Fig. 9e). For other values of $\delta$, the lattice has a broken mirror symmetry as its unit cell cannot overlap with its mirror image. We define the handedness of the parallelogram as CW for $\delta \in (0, 0.5)$ and CCW for $\delta \in (0.5, 1)$ (Fig. 9c, d). $D_{\text{eff}}(\delta)$ for CAPs are qualitatively similar at different $\bar{r}$ values (Fig. S12). Figure 9f shows that $D_{\text{eff}}$ for CW and CCW particles are mirror-symmetric about $\delta = 0.5$ because lattices with the

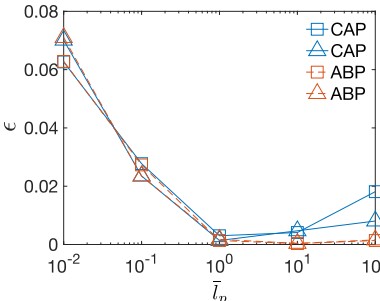

**Fig. 8 | The strength of direction locking ε.** ε defined in Eq. (5) is measured from Fig. 7a–d for CAPs (blue) and ABPs (orange) in square lattice (squares) and triangular lattice (triangles) at a fixed packing fraction $\phi = 0.7$. CAPs undergo reentrant directional locking when $\bar{l}_p > 1$.

same $|\delta - 0.5|$ are mirror images with the same magnitude of chirality. Thus, Fig. 9g only shows $\Delta D_{\text{eff}} = D_{\text{CW}} - D_{\text{CCW}} \in [0, 0.5]$. $\Delta D_{\text{eff}} = 0$ only when the lattice preserves the mirror symmetry at $\delta = 0, 0.5$ and 1, as expected. The maximum difference in $D_{\text{eff}}$ appears at $\delta = 0.3$ for $\bar{r} = 1.5$ (Fig. 9g). CW particles diffuse faster in the CCW lattice ($\delta \in [0.5, 1)$) than in the mirrored CW lattice, namely, a CW lattice with the same value of $|\delta - 0.5|$. Likewise, CCW particles diffuse faster in the CW lattice. Interestingly, these findings are similar to an existing simulation work[83], which found that "levogyre" (CCW) CAPs are trapped in a "left-chiral" flower of ellipse obstacles but can freely enter and leave a "right-chiral" flower, whereas "dextrogyre" (CW) CAPs are trapped in a "right-chiral" flower but can freely enter and leave a "left-chiral" flower. The two works are different for the following reasons: (1) The obstacle particle in ref. 83 is anisotropic, but it is isotropic in our work; (2) the obstacle chirality in ref. 83 is introduced through relative orientations of the ellipse particles, but is imposed through the obstacle

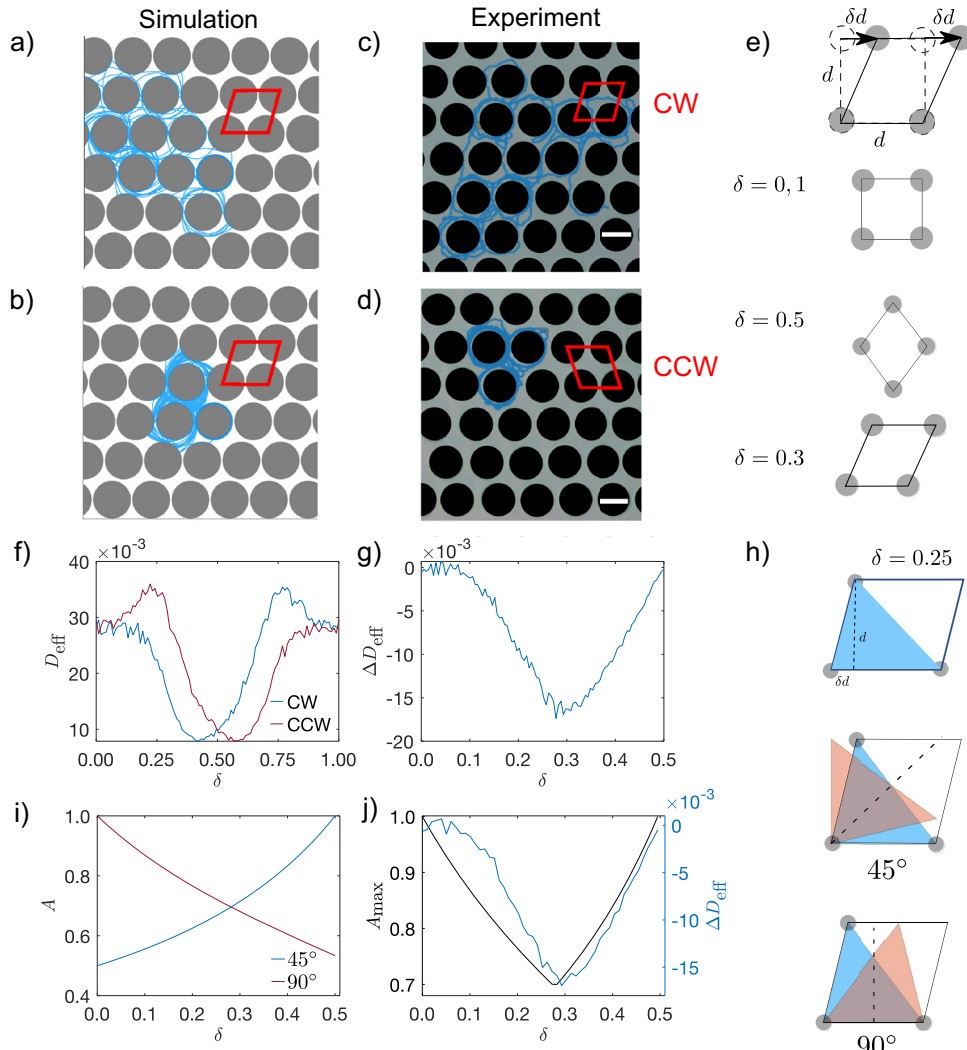

**Fig. 9 | Separation of oppositely handed CAPs by using a mirror-symmetry-broken obstacle lattice.** Simulation trajectories (blue) of CCW (**a**) and CW (**b**) particles in a CW parallelogram lattice with $\delta = 0.3$ and packing fraction $\phi = 0.7$. Experimental trajectories of the grass seed (blue) moving in parallelogram lattices with $\delta = 0.3$ (**c**) and 0.7 (**d**) at $\phi = 0.7$. Scale bars: 1 cm. **e** The parallelogram unit cell can be viewed as a square deformed by $\delta d$. Only $\delta = 0, 0.5$ and 1 preserve the mirror symmetry of the lattice. **f** $D_{\text{eff}}$ of CW and CCW CAPs in parallelogram lattices with

different $\delta$. **g** Difference in effective diffusivity $\Delta D_{\text{eff}}$ over $\delta$. **h** The original blue triangle overlapping with its mirror image (orange triangle). Mirror reflection axes along 45° (middle) and 90° (bottom) and their different overlapping areas. **i** Overlapping area $A$ along 45° and 90° of a unit cell. **j** Strong correlation between $\Delta D_{\text{eff}}(\delta)$ and $A_{\text{max}}(\delta)$. $\bar{r} = 1.5$ and $\bar{l}_p = 100$ for CAPs simulations reported in (**a**, **b**, **f**, **g** and **j**).

configuration here; (3) particles can be trapped in ref. 83, but are diffusive in the obstacle lattice considered here.

To test our simulation prediction, in the experiment we compare the motions of the same CW seed particle with $\bar{r} \approx 1.5$ in the CW ($\delta = 0.3$) and CCW ($\delta = 0.7$) parallelogram lattice (Fig. 9c, d). This comparison is equivalent to comparing the motions of CW and CCW particles in the same parallelogram lattice with $\delta = 0.3$. The experiment confirms our simulation results that CW particles diffuse faster in the CCW lattice than in the CW lattice (Fig. 9c, d and Supplementary Videos 8 and 9). Our further estimate of particle diffusivity indicates that the diffusivity of the seed particle in the two lattices differs by a factor of ~2.7, comparing to ~2.2 in the simulation prediction, showing good agreement between the experiment and simulation.

To understand the differentiation of the two oppositely handed CAPs by the obstacle lattice more quantitatively, we introduce the degree of lattice asymmetry, $A_{max}$, as the fraction of the maximum possible overlapping area $A$ of the unit cell of the lattice with its mirror image. This is motivated by the fact that for a mirror symmetric lattice, the unit cell can overlap with its mirror image via proper translation and rotation; therefore, the maximum possible overlapping area between a unit cell and its mirror image can be used to gauge how much the lattice is away from mirror symmetry. Here we focus on the triangular region delineated within the parallelogram (Fig. 9h). The overlapping area $A$ between the triangular region and the mirror image is computed with respect to two reflecting axes: one along 45° ($A_{45}$) and the other along 90° ($A_{90}$) (Fig. 9h). The degree of lattice asymmetry $A_{max}(\delta) = \max(A_{45}, A_{90})$ is then defined as the maximum value of $A$ obtained from the two aforementioned reflecting axes (Fig. 9i, SI). When $\delta = 0, 1$ or $0.5$ (Fig. 9e), the unit cell has two symmetry axes and can overlap with its mirror image, leading to $A_{max} = 1$ (Fig. 9h). For the other values of $\delta$, $0 < A_{max} < 1$ (Supplementary Videos 10 and 11). Interestingly, $A_{max}(\delta)$ reaches its minimum value when $\delta \approx 0.3$, consistent with the fact that $\Delta D_{eff}$ reaches its minimum at $\delta \approx 0.3$. Moreover, $\Delta D_{eff}(\delta)$ is highly correlated with the geometric parameter $A_{max}(\delta)$ (Fig. 9j), highlighting that the differentiation effect of the obstacle lattice on oppositely handed CAPs, a physical effect, can be quantitatively estimated by the degree of the asymmetry of the lattice, a pure geometric characteristic.

## Discussion

Our simulations show the novel behavior of individual CAPs in the lattices of circular disk obstacles under three cases. The key results are confirmed by granular experiments. Case (1) is about CAP diffusion in the square and triangular lattices. Diffusivity is sensitive to the ratio of the orbital radius of its circular trajectory to the obstacle lattice constant as well as the lattice symmetry. In particular, a CAP can diffuse rapidly in the square lattice but can be caged in the triangular lattice for a long time with the same obstacle packing fraction. A different CAP can exhibit opposite diffusive behavior in these lattices. These results reveal a new way to separate chiral particles by passive lattices, in contrast to the traditional sorting methods of ABPs and CAPs by applying an external field[8,84,86]. A recent simulation study reported the enhanced diffusivity due to the angular speed heterogeneity of CAPs[68]. Our observed enhanced CAPs diffusion by an obstacle lattice can fit into this finding because those obstacles in our system can be regarded as a different type of CAP with zero angular velocity, thereby enhancing particle heterogeneity of the CAPs. Additionally, a recent experimental study has shown that non-tumbling E. coli exhibits similar geometry-sensitive effects[92]. They demonstrated anomalous bacteria size dependent active transport in square lattices, switching from a trapping dominated state for short bacteria to a much more dispersive state for long bacteria. In their experiment, circular motions by short bacteria have hydrodynamic origins and show no intrinsic chirality (i.e., the motion is bidirectional), in contrast to our dry active particle

system in which CAPs have intrinsic chirality and their trapping is due to collisions with obstacles.

In terms of theoretical aspects, a recent analytical work claims that collisions can unexpectedly enhance self-diffusion of chiral active particles in low density regimes[93,94]. This finding showcases a similar behavior to our results. However, it should be noted that our system is operated at high packing fractions and is highly sensitive to the spatial configuration of the obstacles, not merely the lattice density. Therefore, a complete theoretical description of anomalous diffusions of CAPs remains an open question.

In case (2), individual CAPs are subjected to a constant force field. When the reduced persistence length $\bar{l}_p$ is small, CAPs and ABPs are similarly locked along the symmetry axes of the obstacle lattice. As $\bar{l}_p$ increases, the strength of the directional locking effect monotonically decreases to 0 for ABPs but decreases first and then increases for CAPs. In this regime, the intrinsic circular motion of particles is scattered by the lattice, leading to a zigzag motion along a new direction.

In case (3), we propose that a lattice without mirror symmetry has an inherent chirality, which combines with the chirality of the particle; thus, CW and CCW particles with opposite chirality have distinct coupling with the lattice and exhibit different levels of diffusivity. Consequently, the parallelogram lattices of circular obstacles can separate CW and CCW particles, in contrast to previous particle separation methods using non-spherical obstacles[83,91]. We propose using the parameter $A_{max}$ to quantify the degree of lattice asymmetry, which strongly correlates with the diffusivity difference. Hence, the separation of CW and CCW particles can be quantitatively understood by the purely geometric characteristic of the lattice asymmetry.

Overall, our work has examined the interplay of the activity and chirality of particles and the geometry and symmetry of obstacle lattices. Particle diffusivity and migration in obstacle lattices are highly sensitive to the lattice geometry. These features are not found in achiral active particles. Our studies imply that the chiral motions of active particles can sense lattice configurations and this property is unique in CAPs and is absent in ABPs. Thus, CAPs can be used to probe the geometry of an obstacle lattice and other complicated environments. Moreover, we expect that our work can facilitate the applications of CAPs in chemical sensing, separation and therapeutic delivery[41,69,83,91,95].

## Methods

*Echinochloa crus-galli* seeds are purchased from Yimutian inc. In each experimental trial, one seed is placed on a frictional flat cardboard glued with an array of circular plastic disks as obstacles (SI). The cardboard is tightly attached onto a vibrational stage and vibrates vertically at 85 Hz with an amplitude of 0.07 mm. The seed motion is recorded by a camera (Figs. 1d and S2). The position and orientation of the seed in each video frame is identified by openCV[96].

The rotational diffusion coefficient defined in Eq. (1b), $D_r = \lim_{t \to \infty} \langle |\Delta \bar{\theta}|^2 \rangle / (2t)$, where $\langle |\Delta \bar{\theta}|^2 \rangle = \langle |\bar{\theta}(t_0 + t) - \bar{\theta}(t_0)|^2 \rangle_{t_0}$ is the mean squared displacement (MSD) of its angular position subtracted from a constant angular velocity, that is, $\Delta \bar{\theta}(t) = \theta(t_0 + t) - \theta(t) - \omega_0 t$. $\langle \rangle_{t_0}$ represents the average over all $t_0$.

## Data availability

The data generated in this study have been deposited in the Code Ocean repository https://codeocean.com/capsule/8249117/tree.

## Code availability

The codes used in this study have been deposited in the Code Ocean repository https://codeocean.com/capsule/8249117/tree.

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

## Acknowledgements

R.Z. acknowledges support from the Hong Kong RGC grant no. 26302320 and the HKUST Central High-Performance Computing Cluster (HPC3). Y.H. acknowledges support from the Hong Kong RGC grant no. 16305822. The authors also thank Sujit S. Datta for sharing valuable papers and Cynthia Olson Reichhardt for inspiring discussions.

## Author contributions

R.Z. and Y.H. conceived the research. C.W.C. and K.L.F. developed the simulation code and performed the simulations. D.W. and C.W.C. conducted the experiments. K.Q. developed the video analysis code. C.W.C. and D.W. performed data analysis. R.Z., Z.Y., and Y.H. supervised the research. C.W.C., Y.H., and R.Z. wrote the manuscript.

## Competing interests

The authors declare no competing interests.
