## [Peer Review File · Nature Communications]

REVIEWER COMMENTS

Reviewer #1 (Remarks to the Author):

This paper studies the motion of chiral and achiral active particles in 2D lattices on a vibrating surface through physical experiments, and further support their results through simulations. The chiral active particles are two types of seeds (one peeled and another unpeeled) and a 3D printed particle. The authors demonstrate that the diffusivity of the different types of particles can be controlled through the particle itself and the geometry of the objects' placements. They also showed that a specific type of obstacle layout could be used to separate the chiral particles. The authors characterize the behaviors of the different seeds across a parameter space of the persistence length and the orbital radius. I believe the results are interesting and this study is off to a great start but I hope the authors can address my comments so the study's clarity and impact can be increased.

- There are some minor grammar mistakes in some places throughout the text, please make sure to check carefully since this distracts from the essence of the paper.
- Please fix vibration in Fig. 1d to be spelled correctly.
- It might be better to refrain from using the word discover in the introduction (first sentence of last paragraph); some readers might not like this. You are exploiting some of the properties of grass seeds and others have not done that before.
- Right now, the introduction seems to be lacking in the context of potential applications of this work and an explanation of why this work is interesting to a very diverse group of researchers. The following sentence makes it seem like the work is limited and it does not do it justice: "However, experiment on the interplay between chiral active matter and environmental geometry have been scarcely conducted." For Nature Communications it is not enough that others have not explored this route too much. I think it would be much more compelling if you can state why researchers have not expanded this field of research as much (i.e. what are the big challenges here), how is your platform addressing how the environment can be exploited, and what are the possible implications of your work and further research along this route?
- For my comment above, it might be worth mentioning several works (especially along the route of robot swarms and microrobot collectives) where some initial studies have demonstrated how the environment can be exploited to affect the collective's or the swarm's behavior.
- I do not really understand the concept of the mirror-symmetry-broken obstacle lattice. Specifically, could the authors please explain why they compute A_{\max} the way they do. I might be missing something that the authors have already included, but it is not straightforward why this parameter helps to measure how asymmetric the lattice is.
- For a more general audience, it might be interesting to include an experiment with two types of lattice geometries to show in one experiment how the trajectory of a particle changes when it enters a specific section of the lattice. The left half could be a square lattice and the right half could be a

triangular lattice; the grass seed could be placed in one section of the lattice and we could observe what happens when it leaves that section of the lattice and enters the other section. Although this could be a future study in itself, do the authors think a specific trajectory could be designed for these grass seeds just by designing the placement of the obstacle? (Meaning the grass seed goes from start point A, along a specific trajectory, and finally ends up at end point B) This could be a discussion of just a few sentences in addition to the new experiment, but it could help fuel future interesting research questions along this route.

- It would be interesting to see the trajectory of a grass seed within an irregular environment, meaning the obstacles are not in an organized lattice but instead have little variations in their neighbor spacing. This could generate different (maybe more disorganized paths) and it would help us learn more about some of the constraints for these grass seeds.
- My last two comments could be implemented and could be included towards the end of the paper to help show the interesting research questions that this work is opening. I imagine that the microrobotics community would find this type of demonstration interesting and thus it could increase the paper's impact.

Reviewer #2 (Remarks to the Author):

The manuscript entitled "Chiral Active Particles are Sensitive Reporter to Environmental Geometry" investigates the 2D motion of chiral active particles in presence of disk-like obstacles arranged in periodic lattices. To this end, the authors perform experiments using grass seeds that exhibit orbiting motion on a vibrating stage, whose chirality depends on the grooves on the seed surface. The authors also carry out comprehensive numerical simulations based on an active-Brownian-particle model that phenomenologically captures the main features of their experimental system, and which allows them to explore different regimes of the particle dynamics by extensively varying parameters such as bare persistence length of the particle, bare orbit radius, the obstacle packing fraction, external flow velocity, and lattice shape. The main general finding of the paper is that, unlike non-chiral active Brownian particles, the motion of the chiral active particles investigated by the authors is highly sensitive to the detailed structure of the obstacle lattice, which can hinder or enhance the effective particle diffusivity in a nontrivial manner. In addition, chirality can also lead to reentrant directional locking with increasing particle persistent length in presence of an external flow, and particle sorting in lattices with symmetry breaking, all of which are absent in the case of non-chiral active particles. The results presented in the paper show in an elegant manner that chirality of a rather simple active system can give rise to novel effects in presence of obstacles, which have in turn potential applications of great relevance in the field of granular active matter. Therefore I think that

this paper could be considered for publication in Nature Communications after the authors have made a revision of their manuscript taking into account the following remarks:

1. The chiral active particle model used in the numerical simulations is overdamped. At first glance this is puzzling since the grass seeds are millimeter-sized and move in air, so in principle inertial effects are not negligible in both their translational and rotational dynamics. The authors should explain why inertial effects do not seem to play an important role in their experimental system, or at least provide an estimate of translational and rotational inertia of the CAPS with respect to the other forces and torques acting on them.

2. It is not clear why the authors use the simulation units listed in Table 1. Do they match up with the physical units of the experiments upon re-scaling by the effective radius R ? Consistency between the physical and simulation units is important for a quantitative comparison between the different behaviors observed in the experiments and those predicted by the numerical simulations.

3. The authors mention that the second term on the right hand side of equation 1a is a force that embodies the interaction between the CAPS and the obstacles. However, looking carefully at equations 1a and 3 it is not a force but a velocity, i.e. a force divided by some friction coefficient, resulting from that interaction. This must be corrected in the manuscript. It would also be useful to know the value of the characteristic friction coefficient of the grass seeds since this will also allow one to judge whether inertial effects are negligible or not with respect to frictional forces.

4. In the last paragraph of the "Abnormal diffusion" subsection and figures 6b-6e, the authors briefly discuss the behavior of two quantities that are never defined in the manuscript: the number of reversible motion (N) and the ratio of sliding and hopping times (μ). The authors must explain carefully how these two quantities are determined in order to better understand their positive correlation with the effective particle diffusivity.

5. Did the authors carry out any experiment using the vibrated grass seeds to verify the chirality-mediated directional locking in presence of an external field? From figure 7 it is not clear whether these are only numerical simulations.

We sincerely thank the Referees for their time and careful evaluation of our manuscript. Following their questions and comments, we have revised our manuscript and highlighted all the changes in blue. In the following we provide a point-by-point response to the Referee reports. The original Referee comments are shown in *italicized blue*, and our responses are in black. We hope the Referees will find our revised manuscript satisfactory.

Response to Referee 1:

This paper studies the motion of chiral and achiral active particles in 2D lattices on a vibrating surface through physical experiments, and further support their results through simulations. The chiral active particles are two types of seeds (one peeled and another unpeeled) and a 3D printed particle. The authors demonstrate that the diffusivity of the different types of particles can be controlled through the particle itself and the geometry of the objects' placements. They also showed that a specific type of obstacle layout could be used to separate the chiral particles. The authors characterize the behaviors of the different seeds across a parameter space of the persistence length and the orbital radius. I believe the results are interesting and this study is off to a great start but I hope the authors can address my comments so the study's clarity and impact can be increased.

We thank the Referee for the positive and constructive feedback.

- *There are some minor grammar mistakes in some places throughout the text, please make sure to check carefully since this distracts from the essence of the paper. Please fix vibration in Fig. 1d to be spelled correctly.*

We thank the Referee for this comment. We have carefully reviewed and corrected all the grammatical errors in the revised manuscript and marked them blue.

- *Right now, the introduction seems to be lacking in the context of potential applications of this work and an explanation of why this work is interesting to a very diverse group of researchers. The following sentence makes it seem like the work is limited and it does not do it justice: "However, experiment on the interplay between chiral active matter and environmental geometry have been scarcely conducted." For Nature Communications it is not enough that others have not explored this route too much. I think it would be much more compelling if you can state why researchers have not expanded this field of research as much (i.e. what are the big challenges here), how is your platform addressing how the environment can be exploited, and what are the possible implications of your work and further research along this route?*

We thank the Referee for this comment. To better convey the impact and key implications of our research, we have modified our main text as below:

Main text on Page 3–4 Line 55–79:

"Many biological systems are often intrinsically chiral. Experiments on chiral active matter interacting with complex environments are mainly focused on living systems such as bacteria (77–80). Our current understanding of these chiral active entities transport in complex biological environment is overwhelmed by the intricacy of the system involving unclear

physical and biochemical factors (55,65,81). In addition, there is a recent interest in active spinner systems due to their odd viscosities and topological edge currents (63–65). However, it remains unclear how this type of active matter is related to other active systems such as the more commonly studied linear active matter.

In this work, we introduce a new type of granular CAPs, namely, grass seeds, to tackle the above scientific questions. These seed particles are smaller and lighter than previous man-made granular CAPs and are thus suitable for studying CAP motions in obstacle arrays. We investigate how CAPs are transported in obstacle arrays through active Brownian dynamics simulations and granular experiments. Our work provides a simple and convenient platform to study the interplay between chiral active matter and complex environments. The CAPs considered in this work combine the properties of linear propulsion and self-spinning, thus serving as a flexible system to bridge two distinct active matter systems, namely linear and chiral active matter. In contrast to existing works that focus on separating particles with different chiralities using a specific obstacle lattice (82–85), here, we vary lattice parameters and particle chirality and observe novel effects in three systems: (1) abnormal diffusion in periodic lattices, (2) chirality-mediated directional locking in a periodic lattice with an external field, and (3) effective diffusivity difference in lattices without mirror symmetry. Our work reveals that chiral active matter is sensitive to the environmental geometry, including obstacle lattice configuration and the degree of lattice asymmetry. Beyond the commonly discussed applications of active matter in separation and therapeutic delivery, our work also paves the way towards its novel applications such as using chiral active matter as environmental sensors.”

- *For my comment above, it might be worth mentioning several works (especially along the route of robot swarms and microrobot collectives) where some initial studies have demonstrated how the environment can be exploited to affect the collective’s or the swarm’s behavior.*

Following the Referee’s suggestion, we have incorporated the following discussion on these works in the introduction section as your suggestion, including Rubenstein et. al., *Science* (2014), Slavkov et. al., *Sci. Robot.* (2018), Yu et. al., *Nat. Comm.* (2019), Schmidt et. al., *Nat Comm.* (2020), Zhang et. al., *Sci. Robot.* (2021), Wang et. al. *Sci. Adv.* (2021), Wang et. al., *Phys. Rev. Lett.* (2021), Gardi et. al., *Nat. Comm.* (2022), and Chvykov et al., *Science* (2021).

Main text on Page 2 line 28–37:

“...Small robots are a typical example of dry systems, which offers valuable insights into the collective behaviors of active matter in response to their environments. Robot swarms, which navigate, sense, and interact with their environments, demonstrate morphogenesis and on-demand reconfiguration to perform various functions against their surroundings (26-29). Control and actuation of robot swarms often relies on external fields. In particular, magnetic fields have been widely adopted, which enables wireless actuation in complex bio-fluids and showcases significant potential in biomedical applications like targeted drug delivery (30-32). Moreover, these active robot swarms can effectively mimic collective behaviors observed in ecological systems, therefore enabling physical modelling of evolving systems (33).”

- *It might be better to refrain from using the word discover in the introduction (first sentence of last paragraph); some readers might not like this. You are exploiting some of the properties of grass seeds and others have not done that before.*

Following the Referee’s suggestion, we have rephrased relevant sentences in the revised manuscript to avoid using “discover”. The following are the changes we have made:

Main text on Page 3 Line 63:

, we discover a new type of granular CAPs, → , we **introduce** a new type of granular CAPs,

Main text on Page 4 Line 73:

... and discover novel effects on... → ... and **observe** novel effects on ...

Main text on Page 7 Line 140:

We discover two types of abnormal diffusion... → We **observe** two types of abnormal diffusion.

- *For a more general audience, it might be interesting to include an experiment with two types of lattice geometries to show in one experiment how the trajectory of a particle changes when it enters a specific section of the lattice. The left half could be a square lattice and the right half could be a triangular lattice; the grass seed could be placed in one section of the lattice and we could observe what happens when it leaves that section of the lattice and enters the other section. Although this could be a future study in itself, do the authors think a specific trajectory could be designed for these grass seeds just by designing the placement of the obstacle? (Meaning the grass seed goes from start point A, along a specific trajectory, and finally ends up at end point B) This could be a discussion of just a few sentences in addition to the new experiment, but it could help fuel future interesting research questions along this route.*

We thank the Referee for this inspiring suggestion. We make two points over this comment. One point is that we believe that it is possible to design a lattice in which active particles can in theory follow a specific trajectory to move from point A to B. Directional locking effect is a possible candidate for the mechanism. For a longer particle trajectory, diffusion noise can make the task harder to achieve. The other point is that when different lattice structures with considerably different diffusivities are present, we would expect a topotaxis effect, wherein particles would tend to migrate from a high-diffusivity region to a low-diffusivity region [Schakenraad et. al., *Phys. Rev. E* (2022) and Novikova et. al., *Phys. Rev. Lett.* (2017)]. We have performed additional simulations of chiral active particles self-propelling in a binary obstacle lattice, which consists of a square lattice on the left and a triangular lattice on the right (see Fig. S8 below). Using periodic boundary conditions, the initially randomly positioned non-interacting particles quickly move into the square lattice within which they diffuse slower. This demonstrates that topotaxis effect is also present in CAPs. In the experiment, the lack of the periodic boundary condition requires either a long experimental time or many trials, making it difficult to arrive at a clear conclusion. We therefore choose to report on only simulation results regarding this point and leave a systematic study on mixed lattices in the future work.

We have added the above discussion to the revised main text and added Fig. S7 and the following discussion to the revised SI:

Main text on Page 9 Line 176–180:

“Note that when the obstacle lattice is imperfect, interesting transport phenomena may emerge. For example, if an obstacle lattice consists of two different perfect lattices with different diffusivities, particles will exhibit topotaxis effect by migrating into lower-diffusivity region

(10,47). Our additional simulations demonstrate that topotaxis effect is also present for CAPs moving in a binary lattice with identical packing fraction (Fig. S7) ...”

Main text on Page 12 Line 253–256:

“...Note that the directional locking effect can be a candidate mechanism to design a mixed obstacle lattice to guide a CAP along a specific trajectory. However, due to the presence of diffusion noise, the success of this task will become increasingly challenging as the prescribed trajectory lengthens.”

SI on Page 16 Fig. S7:

Figure S7. **Topotaxis in a binary obstacle lattice.** (a) The square lattice on the left and the triangular lattice on the right with $\bar{r} = 1$, $\bar{l}_p = 100$ and the same packing fraction $\phi = 0.6$. The blue line indicates their boundary. (b) Snapshots of 500 non-interacting CAPs (red dot) show a topotaxis effect, i.e., migrating from the high-diffusivity triangular region ($D_{\text{eff}} = 0.15$) to the low-diffusivity square region ($D_{\text{eff}} = 0.02$). Particles are randomly placed at $t = 0$. (c) The evolution of the number of CAPs in the triangular (purple) and square (green) lattices.

SI on Page 6–7 Line 95–104:

“Topotaxis

We observed a topotaxis effect in a binary lattice consisting of a square lattice on the left and a triangular lattice on the right, both with the identical packing fraction of $\phi = 0.6$. Using a periodic boundary condition for both dimensions, we simulated 500 non-interacting CAPs with $\bar{r} = 1$ and $\bar{l}_p = 100$. The simulation results demonstrate that the initially randomly placed CAPs tend to migrate from the high-diffusivity triangular region ($D_{\text{eff}} = 0.15$) to the low-diffusivity square region ($D_{\text{eff}} = 0.02$) (Fig. S7). In the experiment, the lack of the periodic boundary condition requires either a long experimental time or many trials, making it difficult to arrive at a clear conclusion. We therefore choose to report on only simulation results regarding this point and leave a systematic study on mixed lattices in the future work.”

- *It would be interesting to see the trajectory of a grass seed within an irregular environment, meaning the obstacles are not in an organized lattice but instead have little variations in their neighbor spacing. This could generate different (maybe more disorganized paths) and it would help us learn more about some of the constraints for these grass seeds.*

We thank the Referee for this interesting suggestion. We have performed additional simulations and experiments. To quantify the randomness of a slightly irregular lattice, we introduce a disorder parameter via the spatial variance of the lattice:

$$\sigma^2 = \frac{1}{R^2 N} \sum_{i=1}^N |\vec{r}_i - \vec{r}_{0i}|^2,$$

where \vec{r}_i represents the position vector of the obstacle i and \vec{r}_{0i} represents its “equilibrium” position vector in the perfect lattice. We therefore expect that the diffusivity in this perturbed lattice would deviate from that in the reference lattice and behaves intermediate between the diffusivities in perfect triangular and square lattices of the same packing density. Our simulation and experimental results are consistent with this expectation. We have added a brief discussion to the revised main text, and Fig. S8 and the above discussion to **Page 6 Line 78–94** in the revised SI:

Main text on Page 9 Line 180–182:

“...If a single lattice of obstacles is subject to a noisy configuration, CAP diffusivity will deviate from that of the reference perfect lattice; more details are provided in SI (Fig. S8).”

SI on Page 17 Fig. S8:

Figure S8. **CAP diffusion in ordered and disordered lattices with $\phi = 0.6$.** (a) Experimental MSDs in ordered (c and d) and disordered (e) lattices. (b) Simulated MSDs for CAPs with $\bar{r} = 1$ and $\bar{l}_p = 100$ in different obstacle lattices. The irregular lattices used in (a) has variation 0.2. The diffusivity deviates from that in a square lattice and approaches that in a triangular lattice as the degree of disorder σ^2 increases. (c–e) Trajectories of seed 2 in ordered and disordered obstacle arrays for a duration of 60 s. $\sigma^2 = 0.2$ in (e). The trajectory in the irregular lattice exhibits caging in the square lattice and fast diffusion in the triangular lattice. Scale bars: 1 cm.

- *I do not really understand the concept of the mirror-symmetry-broken obstacle lattice. Specifically, could the authors please explain why they compute A_{\max} the way they do. I might be missing something that the authors have already included, but it is not straightforward why this parameter helps to measure how asymmetric the lattice is.*

We thank the Referee for giving us an opportunity to clarify the introduction of A_{\max} . An object is said to have mirror symmetry or reflection symmetry if there exists a mirror-symmetry axis. In our work, we start with a square lattice having mirror symmetry. To break the mirror symmetry, we deform the lattice into a parallelogram lattice by shifting each layer of the obstacles with respect to the layer below by a distance of δd , where $\delta \in [0, 1]$ and d is the lattice constant (see Fig. 9e for illustration and Fig. 9a–d for simulation and experimental examples). Note that when $\delta = 0, 0.5$, and 1, the parallelogram lattice retains mirror symmetry (Fig. 9e) because the unit cell has either a square ($\delta = 0, 1$) or a diamond shape ($\delta = 0.5$). This means that δ is not a good measure of the mirror asymmetry of the lattice. Inspired by the fact that a mirror symmetric object can overlap with its mirror image through proper translation and rotation, we choose to gauge how much a lattice is away from mirror symmetry by measuring

how much its unit cell can overlap with its mirror image. This motivates us to introduce A_{\max} , the maximum possible overlapping area between the unit cell and its mirror image normalized by its area, to characterize the degree of mirror asymmetry. To better convey the above point, we have modified the main text and the SI detailed in below:

Main text on Page 13 Line 267–268:

“For other values of δ , the lattice has a broken mirror symmetry as its unit cell cannot overlap with its mirror image.”

Main text on Page 14 Line 296–300:

“This is motivated by the fact that for a mirror symmetric lattice, the unit cell can overlap with its mirror image via proper translation and rotation after proper translation and rotation; therefore, the maximum possible overlapping area between a unit cell and its mirror image can be used to gauge how much the lattice is away from mirror symmetry.”

SI on Page 7–8 Line 112–121:

“Degree of Lattice Asymmetry

The concept of mirror symmetry inspires us to measure the degree of lattice asymmetry by looking at how much the unit cell of a lattice can overlap with its mirror image. Specifically, to quantify the degree of asymmetry for a parallelogram lattice considered in this work, we measure the maximum possible overlapping area between the unit cell and its mirror (reflected) image along the 45° and 90° axes. This is achieved by calculating the overlapping areas, denoted as A_{45} and A_{90} , which are normalized by the area of the original cell. In particular, $A_{45} = 1$ ($A_{90} = 1$) implies that the image exhibits mirror symmetry along the 45° (90°) axes. Smaller values of A_{45} and A_{90} indicate a higher degree of asymmetry in the structure. We further use the greater value of the two to define the maximum overlapping area $A_{\max} = \max(A_{45}, A_{90})$.”

Response to Referee 2:

The manuscript entitled "Chiral Active Particles are Sensitive Reporter to Environmental Geometry" investigates the 2D motion of chiral active particles in presence of disk-like obstacles arranged in periodic lattices. To this end, the authors perform experiments using grass seeds that exhibit orbiting motion on a vibrating stage, whose chirality depends on the grooves on the seed surface. The authors also carry out comprehensive numerical simulations based on an active-Brownian-particle model that phenomenologically captures the main features of their experimental system, and which allows them to explore different regimes of the particle dynamics by extensively varying parameters such as bare persistence length of the particle, bare orbit radius, the obstacle packing fraction, external flow velocity, and lattice shape. The main general finding of the paper is that, unlike non-chiral active Brownian particles, the motion of the chiral active particles investigated by the authors is highly sensitive to the detailed structure of the obstacle lattice, which can hinder or enhance the effective particle diffusivity in a nontrivial manner. In addition, chirality can also lead to reentrant

directional locking with increasing particle persistent length in presence of an external flow, and particle sorting in lattices with symmetry breaking, all of which are absent in the case of non-chiral active particles. The results presented in the paper show in an elegant manner that chirality of a rather simple active system can give rise to novel effects in presence of obstacles, which have in turn potential applications of great relevance in the field of granular active matter. Therefore, I think that this paper could be considered for publication in Nature Communications after the authors have made a revision of their manuscript taking into account the following remarks:

We thank the Referee for the positive and constructive feedback.

- 1. The chiral active particle model used in the numerical simulations is overdamped. At first glance this is puzzling since the grass seeds are millimeter-sized and move in air, so in principle inertial effects are not negligible in both their translational and rotational dynamics. The authors should explain why inertial effects do not seem to play an important role in their experimental system, or at least provide an estimate of translational and rotational inertia of the CAPS with respect to the other forces and torques acting on them.*

We thank the Referee for this question. In our study, we follow the literature by employing an overdamped active Brownian particle model to investigate the dynamics of macroscopic granular grass seeds on a vibrating substrate [Slebers et al., *Sci. Adv.* (2023) and Giomi et al., *Proc. R. Soc.* (2013)]. The vertical collisions between the particle and the substrate give rise to an active Brownian motion like behavior for the particle in the horizontal plane, which is characterized by a self-propelling velocity v_0 , an angular velocity ω_0 , and a rotational diffusion coefficient D_r . Changing the seed mass or vibration strength will effectively change these parameters. Therefore, for the particle dynamics in the horizontal plane, we use the overdamped equation of motion as a minimal model to elucidate the dynamics of grass seed particles. The good match of the MSDs between the experiment and the simulation (Fig. 1a) justifies our choice of the model. For clarity, we have added the following to the revised manuscript:

Main text on Page 5 Line 88–90:

“...Note that the vertical collisions between the CAP and the substrate give rise to an effective active Brownian-motion like behavior for the CAP in the horizontal plane, and the inertial effect is accounted for by the model parameters v_0 , ω_0 , and D_r .”

- 2. The authors mention that the second term on the right hand side of equation 1a is a force that embodies the interaction between the CAPS and the obstacles. However, looking carefully at equations 1a and 3 it is not a force but a velocity, i.e. a force divided by some friction coefficient, resulting from that interaction. This must be corrected in the manuscript. It would also be useful to know the value of the characteristic friction coefficient of the grass seeds since this will also allow one to judge whether inertial effects are negligible or not with respect to frictional forces.*

We thank the Referee for raising these questions. As we have argued, inertia is important in the vertical direction, but can be accounted for in the horizontal plane by an overdamped model, in which the net force acting on the CAPs in the horizontal plane gives rise to a net velocity

instead of an acceleration. To be more precise, we can express the equation of motion of a CAP in the following format:

$$\frac{d\vec{r}}{dt} = v_0\hat{p} + \alpha\vec{F}, \quad (1a)$$

where α is the CAP's mobility parameter and \vec{F} is the force that embodies the interaction between the CAP and the obstacle j :

$$\vec{F}_j = \begin{cases} -\frac{v_0}{\alpha}(\hat{p} \cdot \hat{N}_j)\hat{N}_j, & \text{if } |\vec{r} - \vec{r}_j| \leq R, \\ 0, & \text{otherwise.} \end{cases} \quad (3)$$

The normal component of the velocity that would drive the particles into the wall is cancelled by the above wall force. This wall potential does not depend on the specific value of α , which is only retained for dimensional consistency. Following the Referee's suggestion, the following changes have been made:

Main text on Page 4 Eq (1a), Page 6 Eq (3) and Line 101–106 are updated according to the above discussion.

Main text on Page 5 Line 87–88:

“...The mobility α and force \vec{F} embody the interaction between the particle and the obstacles...”

Following the Referee's suggestion, we have also measured the kinetic friction coefficient μ_k of the seed particles on the substrate using an inclined plane. Changes made in the revised manuscript are the following:

Main text on Page 7 line 130–131:

“In each experimental trial, one seed is placed on a frictional flat cardboard glued with an array of circular plastic disks as obstacles (SI)...”

SI on Page 4–5 line 50–57:

“Kinetic Friction Coefficient Between Seeds and Cardboard Plate

We measure the kinetic friction coefficient μ_k between the seed particle and the substrate using a 30° inclined plane (Fig. S3). The seed slides down without rotation. We can calculate μ_k by

$$\mu_k = \frac{1}{\sqrt{3}} \left(1 - \frac{2a_{\text{seed}}}{g} \right),$$

where a_{seed} is the acceleration of the seed measured by an image analysis, and g is the gravitational constant. By averaging ten trials of experiment, we find $\mu_k = 0.5$, which is close

to the kinetic friction coefficient between two wood (4).”

SI on Page 13 Fig. S3:

Figure S3: **Measurement of friction coefficient.** The free-body diagram of a seed sliding along a 30° inclined plane.

3. *It is not clear why the authors use the simulation units listed in Table 1. Do they match up with the physical units of the experiments upon re-scaling by the effective radius R ? Consistency between the physical and simulation units is important for a quantitative comparison between the different behaviors observed in the experiments and those predicted by the numerical simulations.*

We thank the Referee for this question. Yes, the experiment and simulation are comparable if we rescale all the length scales by the effective radius R . In the model, the dynamics of CAPs are entirely determined by two dimensionless parameters: (1) persistence length $\bar{l}_p = \frac{v_0}{D_r R}$ and (2) orbital radius $\bar{r} = \frac{v_0}{\omega_0 R}$. In Table 1, we list all the physical values of the seed particles and the corresponding dimensionless quantities used in the simulation. In the simulation, we choose $v_0 = R = 1$ as the fundamental units, and then systematically vary the input parameter of \bar{l}_p and \bar{r} , which effectively changes the value of \bar{D}_r and $\bar{\omega}_0$. All the dimensional quantities are the same between the experiment and the simulation. We have made modifications to the main text and to Table 1 for clarifications.

Main text on Page 6 Line 111–115:

“... We use the particle’s self-propelling velocity v_0 and the effective radius R as the basic units and set $v_0 = R = 1$ with the time scale being R/v_0 . Other physical quantities are rescaled by these units, i.e., $\bar{l}_p = l_p/R$, $\bar{r} = r/R$, $\bar{\omega}_0 = 1/\bar{r}$, and $\bar{D}_r = 1/\bar{l}_p$. Noted that the simulation model can be fully described by the two parameters, namely, \bar{l}_p and \bar{r} .”

Main text on Page 5 Table 1:

Table 1: Model parameters for grass seeds 1 and 2 measured in experiments and in simulations. The values in simulation units are adopted in simulation for comparison with the experimental trajectories in Fig. 1a.

	Experiment			Simulation		
	Unit	Seed 1	Seed 2	Seed 1	Seed 2	
Effective radius	R (mm)	7.0 ± 1	7.0 ± 1	R	1.0	1.0
Self-propelling linear velocity	v_0 (mm s ⁻¹)	39 ± 1	9.7 ± 0.1	v_0	1.0	1.0
Orbital radius	r (mm)	15 ± 2	1.6 ± 0.1	$\bar{r} = r/R$	1.7	0.2
Persistence length	l_p (mm)	25 ± 2	3.6 ± 0.2	$\bar{l}_p = l_p/R$	3.2	0.5
Angular velocity	ω_0 (s ⁻¹)	-2.6 ± 0.4	6.3 ± 0.3	$\bar{\omega}_0 = 1/\bar{r}$	0.59	5.0
Orientalional diffusion coefficient	D_r (s ⁻¹)	1.6 ± 0.1	2.7 ± 0.2	$\bar{D}_r = 1/\bar{l}_p$	0.31	2.0

4. *In the last paragraph of the "Abnormal diffusion" subsection and figures 6b-6e, the authors briefly discuss the behavior of two quantities that are never defined in the manuscript: the number of reversible motion (N) and the ratio of sliding and hopping times (μ). The authors must explain carefully how these two quantities are determined in order to better understand their positive correlation with the effective particle diffusivity.*

We thank the Referee for giving us the opportunity to clarify the definitions of N and μ . We have added the following to the revised main text and the SI:

Main text on Page 10 Line 196–209:

“...Specifically, for a fixed duration of simulation t_{total} , we measure the sliding time of a particle, t_{slide} , defined as the cumulative time when CAPs are in contact with the surface of obstacles (i.e., there exists an obstacle i such that $|\vec{r} - \vec{r}_i| \leq R$). Particle hopping time is then defined via $t_{\text{hop}} = t_{\text{total}} - t_{\text{slide}}$. During sliding, we count the number of occurrences of reversible motion of the particle as N (i.e., particle’s tangential velocity with respect to the obstacle surface normal changes sign). Both parameters we have measured show strong positive correlations with Ψ , indicating that they dominate the diffusion constant D_{eff} (Fig. 6).

To understand, note that the diffusion of CAPs is primarily driven by their ability to hop to the next row of obstacles, effectively initiating a new orbit. The occurrence of a larger number of reversible motions on the surface of obstacles provides CAPs with more opportunities to explore the surrounding space. Consequently, this increased exploration enhances the likelihood of CAPs successfully hopping to the next row of obstacles. Notably, our results demonstrate a strong positive correlation between N and the effective diffusivity of CAPs.”

SI on Page 5 Line 59–64:

“Particle hopping time t_{hop} can be calculated as $t_{\text{total}} - t_{\text{slide}}$, representing the total time CAPs spend in hopping between obstacles without contacting surfaces of obstacles. The ratio μ indicates the dominant mode of motion for CAPs. If $\mu > 1$, it suggests that sliding is the dominant mode of motion. Conversely, if $\mu \leq 1$, it indicates that hopping becomes the

dominant mode. Notably, N is proportional to μ . This is because a larger sliding time provides a larger chance for the reversible motion to happen on the surface of obstacles.”

5. *Did the authors carry out any experiment using the vibrated grass seeds to verify the chirality-mediated directional locking in presence of an external field? From figure 7 it is not clear whether these are only numerical simulations.*

We appreciate this comment provided by the Referee. For the directional locking phenomenon, experimental realization is difficult. We tried to apply the external field using gravity. However, the cardboard surface on the inclined plane does not vibrate very uniformly and large noises generated by lifting/tilting the vibration table, which can cause damage to the vibration system. Moreover, vibrating inclined plane can rotate the light seed which is beyond our simulation model. We managed to perform a few trials within a short period, while the directional locking effect is not observed in experiment due to the above problems. With these experimental difficulties, we leave it for the future work.

We have added the above discussion to **Page 8 Line 130–137** in the revised SI to convey the above difficulties in the experiment.

REVIEWERS' COMMENTS

Reviewer #1 (Remarks to the Author):

The authors addressed all of my points and I believe the paper is in very great shape. I recommend this paper for publication in its current form.

Reviewer #2 (Remarks to the Author):

The authors have satisfactorily responded to all my questions and made the necessary changes to the manuscript. Therefore I recommend this paper for publication in Nature Communications in its current form.